# Genome-Wide Identification of *Glutathione S-Transferase* Genes in Eggplant (*Solanum melongena* L.) Reveals Their Potential Role in Anthocyanin Accumulation on the Fruit Peel

**DOI:** 10.3390/ijms25084260

**Published:** 2024-04-11

**Authors:** Hesbon Ochieng Obel, Xiaohui Zhou, Songyu Liu, Yan Yang, Jun Liu, Yong Zhuang

**Affiliations:** 1Institute of Vegetable Crops, Jiangsu Academy of Agricultural Sciences, Nanjing 210014, China; 20230024@jaas.ac.cn (H.O.O.); 20100029@jaas.ac.cn (X.Z.); 20180052@jaas.ac.cn (S.L.); 20170007@jaas.ac.cn (Y.Y.); 20110024@jaas.ac.cn (J.L.); 2Laboratory for Horticultural Crop Genetic Improvement, Nanjing 210014, China

**Keywords:** eggplant, anthocyanin, *glutathione S-transferase*, *SmGSTF1*

## Abstract

Anthocyanins are ubiquitous pigments derived from the phenylpropanoid compound conferring red, purple and blue pigmentations to various organs of horticultural crops. The metabolism of flavonoids in the cytoplasm leads to the biosynthesis of anthocyanin, which is then conveyed to the vacuoles for storage by plant *glutathione S-transferases* (*GST*). Although *GST* is important for transporting anthocyanin in plants, its identification and characterization in eggplant (*Solanum melongena* L.) remains obscure. In this study, a total of 40 *GST* genes were obtained in the eggplant genome and classified into seven distinct chief groups based on the evolutionary relationship with *Arabidopsis thaliana* GST genes. The seven subgroups of eggplant GST genes (*SmGST*) comprise: dehydroascorbate reductase (DHAR), elongation factor 1Bγ (EF1Bγ), Zeta (Z), Theta(T), Phi(F), Tau(U) and tetra-chlorohydroquinone dehalogenase TCHQD. The 40 *GST* genes were unevenly distributed throughout the 10 eggplant chromosomes and were predominantly located in the cytoplasm. Structural gene analysis showed similarity in exons and introns within a *GST* subgroup. Six pairs of both tandem and segmental duplications have been identified, making them the primary factors contributing to the evolution of the *SmGST*. Light-related cis-regulatory elements were dominant, followed by stress-related and hormone-responsive elements. The syntenic analysis of orthologous genes indicated that eggplant, Arabidopsis and tomato (*Solanum lycopersicum* L.) counterpart genes seemed to be derived from a common ancestry. RNA-seq data analyses showed high expression of 13 SmGST genes with *SmGSTF1* being glaringly upregulated on the peel of purple eggplant but showed no or low expression on eggplant varieties with green or white peel. Subsequently, *SmGSTF1* had a strong positive correlation with anthocyanin content and with anthocyanin structural genes like *SmUFGT* (*r* = 0.9), *SmANS* (*r* = 0.85), *SmF3H* (*r* = 0.82) and *SmCHI2* (*r* = 0.7). The suppression of *SmGSTF1* through virus-induced gene silencing (VIGs) resulted in a decrease in anthocyanin on the infiltrated fruit surface. In a nutshell, results from this study established that *SmGSTF1* has the potential of anthocyanin accumulation in eggplant peel and offers viable candidate genes for the improvement of purple eggplant. The comprehensive studies of the *SmGST* family genes provide the foundation for deciphering molecular investigations into the functional analysis of *SmGST* genes in eggplant.

## 1. Introduction

Eggplant (*Solanum melongena* L.) plays a significant role as a vegetable crop by offering both financial benefits and essential nutrients to individuals [1]. Annually, 55 million tonnes of eggplant are produced worldwide, and 35 million tonnes—constituting over 60% of the global eggplant production—are produced in China alone, annually [2]. As an important commercial characteristic, eggplant with a purple fruit peel has gained special attention among researchers due to the increasing need for integrating breeding schemes to develop nutritious vegetables [3,4]. The concentration of anthocyanin and chlorophyll content are integral in determining eggplant peel color.

Flavonoid compounds are common secondary metabolites found in plants that give rise to a diverse group of water-soluble pigments known as anthocyanins. The anthocyanin pigmentations are attributed to the purple coloration in the leaves, flowers, fruit peel and flesh. The degree of purple color intensity and expression is determined by the vacuolar PH levels [5,6]. Anthocyanins are key attractants or repellants to insects and animals [7]. In plants, anthocyanins play a protective role against aberrant stress factors such as offering resistance against biotic and abiotic phenomena including the attack by pathogens and abnormal temperatures caused by ultraviolet radiation and extremely low temperatures [8,9]. Anthocyanins possess strong antioxidant capacities to enhance human health by providing defense against cancer, cardiovascular disease and a variety of other ailments [10]. Several studies in horticultural crops have discussed the anthocyanin biosynthesis and its regulation mechanisms [11,12,13,14].

The biosynthesis of anthocyanin involves a complex phenylpropanoid pathway orchestrated by a myriad of enzymes catalyzing a chain of biosynthetic stages. Phenylalanine, the precursor of the biosynthesis of anthocyanin, is chiefly synthesized in the cytoplasm via the flavonoid metabolic pathway [15,16]. The biosynthesis of anthocyanin involves a group of enzyme complex called cinnamates hydroxylase, flavanone-3b-hydroxylase and flavonoid 3-hydroxylase. These three enzyme complexes fuse onto the endoplasmic reticulum via the soluble subunits. The integrity of anthocyanin biosynthetic enzymes is coordinated by transcription factor complex of MYB, bHLH and WD40 proteins (MBW) [17]. Anthocyanins are not only conveyed to the vacuole by glutathione *S*-transferase but are also mediated by two other transport mechanisms known as vesicle trafficking and membrane transporters [18,19,20].

*Glutathione S-transferases* (*GST*) are important enzymes located downstream of cytochrome P450 and are involved in the transportation of anthocyanin to the vacuoles. They also possess defense systems due to their active antioxidant properties [21,22]. The *GST* superfamily is involved in scavenging different chemical compounds by coupling glutathione (GSH) and reducing it into a hydrophobic compound which in turn makes the compound more hydrophilic and can easily be removed through the vacuole [23]. *GST* plays an integral role in plant morphosis and secondary metabolism [22,24]. *GST* is also involved in signal communication pathways, tetrapyrrole metabolism, retrograde signaling and detoxification of reactive carbonyl species [25,26,27]. *GST* is known, on one hand, to have the N-terminal domain exhibiting the canonical thioredoxin fold distinctly to possess consisting of α-helices and β-elements having a β1α1β2α2β3β4α3 regional anatomy and G-site where glutathione binding occurs. On the other hand, C-terminal domains have the presence of α-helices and H-sites involved in the binding of secondary substrates exhibiting hydrophobicity [22].

The *GST* has three distinct classification systems based on subcellular localization; these include cytoplasmic *GST*, mitochondrial *GST* and microsomal superfamilies *GST*. The soluble *GST* are mainly the cytoplasmic and mitochondrial types, while the microsomal *GST* are membrane-bound proteins participating in the metabolism of eicosanoid and glutathione [28,29]. Soluble plant *GST* are classified into fourteen subfamilies based on their sequence and structural similarity as well as similar functional roles as follows: tetra-chlorohydroquinone dehalogenase (TCHQD), elongation factor 1Bγ (EF1Bγ), microsomal prostaglandin E-synthase type 2 (mPGES2), dehydroascorbate reductase (DHAR), glutathionyl hydroquinone reductase (GHR), iota, hemerythrin, Metaxin, Lambda (L), Zeta (Z), Ure2p, Theta (T), Phi (F), Tau (U). Another type of *GST* classification is on the active site residue, which is either serinyl or cysteinyl *GST* [30,31]. Five subfamilies: TCHQD, Zeta, Theta, Phi and Tau have serinyl (Ser) [32]; and seven other subfamilies: iota Lambda, Metaxin, GHR, DHAR, Hemerythrin and mPGES2 *GST* have cysteinyl (Cys) catalytic residues active sites. The serinyl are dimeric protein, while cysteinyl is a monomeric protein. The cysteinyl and serinyl *GST* perform diverse functions; for example, the serinyl group is involved in the glutathionylation reactions, comprising coupling of glutathione to foreign substances, rendering them soluble and they are subsequently expelled from the plant cell. On the other hand, cysteinyl *GST* facilitates deglutathionylation reactions and also catalyzes the reduction of dehydroascorbate [32,33,34].

Previous reports have unveiled the involvement of some candidate *GST* genes in anthocyanins biosynthesis in different plant species, including: *AtTT19* in *Arabidopsis thaliana* [35], *VviGST1*, *VviGST3* and *VviGST4* in *Vitis vinifera* [36], *LcGST4* in *Litchi chinensis* Sonn [37], *MdGSTU12* in *Malus domestica* [38], *MdGSTT6* in *Malus domestica* [39], *AcGST1* in *Actinidia chinensis* [40], *PsGSTF3* in *Paeonia suffruticosa* [41], *CsGSTF1* in *Camellia chinensis* var purple [42], *PcGST57* in *Pyrus communis* [43]. The above studies depict that GSTs are highly conserved in anthocyanin biosynthetic pathways. Even though comprehensive data on the genome sequence of eggplant is accessible to the public, the systemic analysis and understanding of the genes that encode for the *GST* gene family in eggplants remains unexplored. We performed a comprehensive analysis of the entire genome of eggplant and obtained a total of 40 *GST* members; we then characterized their genetic composition, phylogenetic relations and expression patterns in fruit tissues. The findings herein provide valuable insights into various aspects of eggplant *GST*, including their expression specifically in the peel of eggplant fruit. Furthermore, the functionality of *SmGSTF1* was confirmed through the utilization of virus-induced gene silencing (VIGS), to explore the role of the *SmGSTF1* gene in anthocyanin sequestration in purple eggplant fruit. The study of the eggplant *GST* gene family could contribute significantly to the breeding of anthocyanin-rich eggplant varieties.

## 2. Results

### 2.1. Genome-Wide Identification and Analysis of GST Family Genes in Eggplant

The characteristics of the *GST* family genes and their distinct composition were obtained. Here, 40 *GST* family members were identified following the BlastP sequence search using Arabidopsis protein sequences as the query and hidden Markov model. The *SmGST* family genes were categorized and designated based on the similarity arising from the members of the *GST* family between eggplant and Arabidopsis. The lengths of the *SmGST* genes ranged from 782 bp (*SmGST9*) to 13,407 bp (*SmGSTZ1*), with the fewest amino acids being 78 (*SmGST19*) and the highest being 414 (*SmGST18*). The molecular weights of *SmGST* were between 9.167 KDa (*SmGST19*) and 47.144 KDa (*SmGST18*). Variations of the theoretical isoelectric points were observed to be ranging from 4.4 (*SmGST19*) to 9.45 (*SmGST21*). In terms of the subcellular localization predictions, the *SmGST* was predominantly found in the cytoplasm and partly in the chloroplast and nucleus (Table 1). The instability index ranged from 25.18 to 58.87, with an instability index above 40 considered unstable. The aliphatic index was as low as 78.21 (*SmGST18*) and the highest was 129.74 (*SmGST17*). Hydrophilicity indexes of most *SmGST* genes were mainly negative except for *SmGST1* and *SmGST19*, which were positive, expressing hydrophilicity of the *SmGST* proteins but to varying degrees (Appendix A).

### 2.2. Phylogenetic Analysis of the GST Family Genes

The evolutionary relationship of *GST* family members was performed by comparing them to different plant species. A total of 185 full-length GST protein sequences from three different plant species—eggplant, Arabidopsis and tomato, comprising 40, 64 and 81 GST protein sequences, respectively—were analyzed (Appendix A) by aligning the sequences using MEGA X software version 10.2.5. The identified *SmGST* eggplant proteins belonged to seven subgroups, as follows; Tau subfamily, Zeta subfamily, Lambda subfamily, Theta subfamily, Dehydroascorbate-reductase (DHAR) subfamily, Tetrachlorohydroquinone dehalogenase-like (TCHQD) subfamily, Phi subfamily and elongation factor 1 gamma (EF1Bγ) subfamily. The Tau subfamily emerged as the dominant group, accounting for 62.5% of the total number of *SmGST* proteins. Other subclasses have the following distributions; Phi (*SmGSTF1–4*), Zeta *(SmGSTZ1–2*), Theta (*SmGSTT1–3*), DHAR (*SmDHAR1–2*), TCHQD (*SmTCHQD1–2*), EF1Bγ (*SmEF1 Bγ1–2*) (Figure 1). Evolutionary analysis unveils that the eggplant SmGST proteins exhibited homology with Arabidopsis and tomato proteins, insinuating that the origin and divergence of *GST* genes among these species was conservative and could perform similar physiological processes.

### 2.3. The Phylogeny, Gene Structure and Motif Composition of the SmGST Genes

Based on the phylogenetic relationships, *SmGST* belongs to seven groups: TAU, Zeta, EF1Bγ, DHAR, Theta, TCHQD and Phi (Figure 2A). The analysis of genomic sequences of eggplant *SmGST* genes revealed that all the *SmGST* genes have intron numbers ranging from 1 to 12. A total of 60 percent of the *SmGST* genes contained a single intron (Figure 2B). The Zeta subfamily was found to contain the largest number, with 10 exons and nine introns in *SmGSTZ2,* while *SmGSTZ1* has nine exons and nine introns. *SmGST* genes among the Tau subfamily comprised a pair of exons and a single intron, except *SmGSTU8* which has two introns. The Phi subfamily members contained three exons and one intron each. Most of the *SmGST* genes have an untranslated region (UTR) except for six *SmGST* genes *(SmGSTU17*, *SmGSTU21*, *SmGSTU20*, *SmGSTU6*, *SmDHAR2* and *SmTCHQD2*).

The conserved domain prediction and analysis revealed that 40 eggplant *SmGST* genes contained both the N and C terminus GST domains. Motif analysis identified 10 conserved motifs in the *SmGST* and denoted them as motifs 1–10. Upon analyzing motifs, 1, 3 and 5 identified through annotation belonged to the GST-N domain; while 2, 4 and 6 are identified to be the GST-C domain (Appendix A). Similar motif patterns were obtained among *SmGST* in the same subfamily and are likely to perform similar biological functions. Motif 3 was present in all *SmGST*. The majority of the genes have motif 2; this is an indication that motif 2 acts as *SmGST* identifying markers (Figure 2C). Many of the motif types were found within the Tau subfamily. Description of the motif 1–10 sequences and their functional relevance are provided in Appendix A. Only motif 1 denoting glutathione S-transferase Omega/Tau-like (IPR045073) have a known function, while the others have non-predicted functions.

### 2.4. Distribution of SmGST Genes on the Chromosomes and Their Duplication

The distribution of *SmGST* genes on the eggplant chromosomes was displayed using Tb tools according to the eggplant genomic information. A total of 40 *SmGST* genes manifested non-uniform distribution across the 10 chromosomes (Figure 3A). A total of 12 genes, representing 30% of *SmGST* genes, are distributed on chromosome 9, preceded by six genes on chromosome 10. Other chromosomes had fewer *SmGST* genes as follows: five *SmGST* in chromosome 4; four *SmGST* genes on chromosomes 1 and 5, respectively; three *SmGST* genes on chromosome 6; chromosomes 2 and 8 each possess two *SmGST* genes; while chromosomes 3 and 11 each have a single *SmGST* gene. Nevertheless, chromosomes 7 and 12 did not exhibit any protein distribution. The absence of the *SmGST* genes on some chromosomes might be due to loss or doubling in the event of the occurrence of evolution. Such loss or duplications have no direct linkage with the chromosomal architecture in terms of length, size and other parameters. Gene duplication analysis showed that six pairs of segmental duplicated genes and six pairs of tandem duplication were harbored in the 10 chromosomes but with uneven distribution patterns. Six pairs of tandem replications were obtained, these include: *SmGSTU11*/*SmGSTU12*, *SmGSTU16*/*SmGSTU17*, *SmGSTU23*/*SmGSTU24*, *SmGSTU6*/*SmGSTU7*, *SmGSTZ1*/*SmGSTZ*2 and *SmGSTTT2*/*SmGSTT3;* and six pairs of segmental replications were identified: *SmGSTU9*/*SmGSTU19*, *SmEF1Bγ1*/*SmEF1Bγ2*, *SmDHAR1*/*SmDHAR2*, *SmTCHQD1*/*SmTCHQD2*, *SmGSTF1*/*SmGSTF2* and *SmGSTF3*/*SmGSTF4* (Figure 3B and Appendix A). Blue curves connect the segmentally duplicated genes, while tandems are joined by red curves (Figure 3B).

Further, the Ka and Ks values were calculated through the Ka/Ks calculator in TBtools version 2.042. The Ks values were ranging between 0 and 2.912. In addition, the computation of Ka/Ks values of duplicated *SmGST* genes exhibited values below 1, depicting a robust purifying selection during their evolvement. Duplication of genes occurred millions of years ago (Mya). The duplication phenomena of *SmGST* genes occurred from about 5.879 Mya (Ks = 0.17636) to 97.06 Mya (Ks = 3.399) (Appendix A).

### 2.5. Collinearity Analysis of the SmGST Genes in Eggplant

The collinearity of eggplant *SmGST* genes with the *GST* genes in Arabidopsis and tomato was analyzed using a dual synteny plot for MCScanX in TBtools. The analysis displayed some of the collinearity relationships of *SmGST* genes with those of Arabidopsis and tomato. This is an indication that eggplant exhibited a high degree of conservation among its *SmGST* genes. Arabidopsis showed 13 collinearity blocks versus eggplant while tomato displayed 24 collinear blocks with the eggplant *SmGST* genes (Figure 4, Appendix A). Following the outcome of collinearity, two graphical maps were displayed, signifying genes that are analogous in structure and function due to their common evolutionary origin between eggplant and Arabidopsis, and that of eggplant and tomato, respectively.

### 2.6. Analysis of Cis-Acting Elements in the Promoters of SmGST Genes

Cis-acting regulatory elements of the 40 *SmGST* genes were analyzed using 2000 bp sequences located at the tail end of the initiation of transcription position and were uploaded to the online PlantCARE website. The cis-acting regulatory features were grouped into four categories, as follows: stress-related factors, phytohormones and growth and development-related elements. Light-responsive were the dominant group. Light-related elements manifested in all the *SmGST* genes accounting for 38% of the elements (Figure 5B); this is an indication of the importance of light stimulatory mechanisms to the expression of *SmGST* genes. The detailed hormone-responsive elements which comprised the second largest group (29%) were as follows: MeJA (methyl jasmonate) response (CGTCA and TGACG motifs), abscisic acid response (ABRE), gibberellin response (GARE-motif, P-box and TATC-box), auxin response (AuxRR-core and TGA-elements) and salicylic acid-responsive (TCA-element). Stress-related elements, in the promoter of *SmGST* genes, include anaerobic factors (ARE and GC-motif), low-temperature response (LTR), drought-response (MBS and MYC), and defense and stress (TC-rich repeats) (Figure 5A). Several cis-acting elements associated with plant development were present, including CAT-box, circadian, GCN4_motif, HD-Zip 1, MBSI, MSA-like and O2-site, zein metabolism, meristem expression and endosperm development (Figure 5A). Moreover, *SmGST* promoters also contained several transcription factors binding sites, including the WRKY-binding motif (W box), Myb-binding site, MYB-like sequence and MYB recognition site and can directly control *SmGST* transcriptions and have been shown to have a direct influence on flavonoid biosynthesis. *SmGST* genes depict different compositions and proportions of cis-regulatory elements. *SmDHAR1* has the largest proportion of abiotic stress-responsive elements, while *SmGSTF4*, *SmGSTU1*, *SmGSTU18*, *SmGSTU*9 and *SmTCHQD1* have the largest proportion of light-responsive elements (Appendix A).

### 2.7. Expression Patterns of SmGST Genes in the Peel of Different Eggplant Varieties

The expression of *SmGST* genes was determined using RNA-seq data from our recent research group data [44] (Appendix A). Eggplant cultivars of different peel colors were used, including: cultivars with dark-purple peel—A1; green peel—A2; black-purple with green calyx—A3; white peel cultivar—A4; and cultivars of reddish-purple—A5. Significant expression patterns were observed among the *SmGST* genes in the peel of different eggplant cultivars; this is an indication of distinct physiological functions in eggplant (Figure 6A). High expression levels were recorded in 13 *SmGST* genes (*SmGSTU13*, *SmGSTU24*, *SmGSTU4*, *SmGSTU7*, *SmGSTU12*, *SmGSTZ2*, *SmGSTF1*, *SmGSTU16*, *SmDHAR2*, *SmGSTU11*, *SmGSTU22*, *SmGST18*, *SmGSTU25*). These genes were upregulated on the purple eggplant varieties, thus indicating that they could directly or indirectly influence the mechanisms of anthocyanin accumulation on the peel of eggplant. *SmGSTF1* was glaringly upregulated in purple varieties of black-purple and reddish-purple eggplant peels but was downregulated or not expressed in white (A4; white) and green (A2; green) peel colors.

The information from the RNA-seq data analysis gives evidence for further exploring *SmGST* genes through qRT-PCR analysis. qRT-PCR analysis was applied on nine *SmGST* genes to determine their pattern of expression among five fruit tissues (fruit peel, calyx, fruit calyx, petal and anther) derived from dark-purple eggplant variety, denoted as EP26. The primers used for qRT-PCR analysis are recorded in Appendix A. The qRT-PCR analysis confirmed an elevated expression level of *SmGSTF1* in the peel which corroborated with the RNA-seq quantifications, followed by the petals and anther, in that order (Figure 6B). The result indicates that *SmGSTF1* tended to show high expression levels in various eggplant fruits. Apart from the *SmGSTF1*, the *SmGSTU4* gene also tended to be more highly expressed in the peel, anther, petal and fruit calyx. *SmGSTU4* was the least expressed in all the tissues except in the petal tissue, where *SmGSTZ2* was the least expressed. Notably, the *SmGST* genes tested exhibited high expression in the peel and low expression in the calyx and fruit calyx. Consequently, the expression of the *SmGST* genes was generally higher in dark-purple compared to light-purple eggplant peel in fruit peel, anther and petal tissues; while fruit calyx and calyx showed vice versa, signifying variation of SmGST in different tissues.

To further elucidate the function of GST genes in the peel of eggplant, we performed a correlation analysis between *SmGST* genes, anthocyanin structural genes and anthocyanin content (Figure 6C). *SmGST* genes showed variations in the correlation with anthocyanin structural genes. *SmGSTF1* showed a high positive correlation with anthocyanin structural genes like *SmUFGT* (*r* = 0.9), *SmANS* (*r* = 0.85), *SmF3H* (*r* = 0.82) and *SmCHI2* (*r* = 0.7). Other anthocyanin structural genes *SmFLS* and *SmC4H* also showed high expression with most of the *SmGST* genes. Correlation analysis revealed a significant positive correlation between *SmGSTF1* and total anthocyanin content (*r* = 0.82, *p* = 0.05) (Figure 6C). Finally, the *SmGSTF1* gene was selected for further analysis due to its apparent upregulation on eggplant fruit with purple peel color.

### 2.8. Effects of Silencing SmGSTF1 on the Eggplant Peel Color

As a result of the apparent high expression levels revealed by the combination of RNA-seq and qRT-PCR, the *SmGST* gene was subjected to gene silencing experiments. This aids in determining the functional role of the *SmGST* gene in response to peel color change and anthocyanin pigmentation in the peel of purple eggplant fruit. Compared to eggplant fruit infected with *Agrobacterium tumefaciens*, TRV2, the *SmGSTF1*-induced silenced eggplants showed reduced anthocyanin, causing a pale appearance around the infiltration sites in the fruit peels. Generally, the study indicates that *SmGSTF1* could contribute to anthocyanin biosynthesis in purple eggplant peel (Figure 7).

## 3. Discussion

Anthocyanins are important dietary nutritional pigments to humans; hence, studies focusing on the biosynthesis and accumulation of anthocyanin in horticultural crops are paramount. Anthocyanin biosynthesis occurs in the cytoplasm following the flavonoid metabolic pathways and is stored in the vacuoles [14]. Anthocyanin deposition into the vacuole involves transport machinery like GST, membrane transport, or vesicle movements [5,40]. Higher plants have a GST supergene family with enzymatic reactions and perform diverse functions and secondary metabolic activities. The GST is grouped into different subclasses such as TAU, Phi, Lambda, Zeta, TCHQD, Theta, GHR, EF1B and DHAR based on the previous report [45]. Identification of GST genes in other plant species has been reported; for example, 55 GSTs in Arabidopsis [46], 38 GSTs from the apple (*Malus domestica* [38] and 82 GSTs in radish (*Raphanus sativus*) [47], 90 in potato (*Solanum tuberosum* L.) [48], 90 in tomato [49], 74 in soybean (*Glycine max*) [50], 85 in pepper (*Capsicum annuum*) [51], 92 in grapes (*Vitis vinifera*) [52], 57 in pear (*Pyrus communis* L.) [43] and 59 in tobacco (*Nicotiana tabacum*) [53]. In this study, 40 GSTs were found in the eggplant genome and subsequently characterized (Table 1).

It is hypothesized that all GST genes have a common origin and the duplication of genes could potentially lead to an increase in the number of genes. This is attributed to the amino acid sequence similarity and the genomic positions [54,55]. After conducting a phylogenetic examination of 185 GST proteins sourced from Arabidopsis, tomato and eggplant, the 40 SmGST proteins were divided into seven groups, which was in corroboration with those of radish [47], with Tau classes being the most numerous groups comprising 25 members, followed by the Phi class comprising 4 members (Figure 2A). This corroborates with previous findings that the subfamily Tau and phi are generally the dominant *GST* in plants [56]. The *SmGST* proteins in the same subfamily are likely to possess similar biological functions. Chromosome rearrangements often lead to segmental duplications, resulting in numerous replicated segments of the chromosome. On the other hand, when many members from homologous gene families are found adjacent to each other on a single chromosome, this results in tandem duplication [54]. Duplication events can give rise to a large number of gene families and facilitating their amplification is important in the evolution of species genomes [54]. The expansion of the *GST* gene family is primarily attributed to the enlargement of the TAU and Phi subfamilies. In our study, we observed that the expansion of the *SmGST* family genes was attributed to both the tandem and segmental duplications (Appendix A).

Cis-acting regulatory elements analysis of *SmGST* genes contained light-responsive, stress-related phytohormones and growth-related elements and transcription binding factors related to flavonoid biosynthesis (Figure 6). This suggests that an internal element may be regulating the expression of *SmGST* genes. Differences in gene structure can arise from variations in exon and intron structures [57,58]. The presence of introns can lead to increased evolutionary diversity through alternative splicing and exon shuffling [59]. Individuals belonging to a particular group probably possess comparable genetic composition. Analysis of *SmGST* gene architecture revealed that genes of the same group had similar architecture, indicating a high level of conservation at splicing sites throughout evolution.

The accumulation of anthocyanin in the peel of eggplant fruit has a direct impact on the color of the peel. Fruit color is highly valued in terms of appearance and commercialization. The breeding of eggplants aims to produce fruits with a dark-purple color, which is considered a desirable trait [60]. While the anthocyanin pathway has received extensive explorations and studies have been put forward, there is still a need for further investigation of the transport pathway and mechanisms of *GST* anthocyanin sequestration. The present investigation successfully identified *GST* gene expression following their expression on the peels of eggplant with different phenotypes: dark purple, light purple, red-purple, white and green peel. Analysis of differential gene expression revealed that *SmGSTF1* expression exhibited a notable significant difference between dark purple cultivar and red-purple fruits compared to white and green fruits (Figure 6A). Consequently, the role of *SmGSTF1* in promoting anthocyanin accumulation was further examined and the results confirmed its potential function in this process.

As part of investigating the genes responsible for anthocyanin accumulation in eggplant, expression profiles of *SmGST* genes were analyzed in the fruit peel and fruits of different eggplant varieties (Figure 6A,B). Our findings point to a positive correlation between the expression of *SmGSTF1* and anthocyanin content (Figure 6C). Some of the *SmGST* genes also showed a positive correlation with anthocyanin structural genes, indicating synergistic functions; while others have a negative correlation depicting antagonistic or different roles. In our previous research group, transcriptomic analysis [44] explored the expression of anthocyanin structural biosynthetic genes in the five different eggplant varieties. The majority of the structural genes involved in flavonoid biosynthesisincluding *PAL*, *C4H*, *4CL*, *CHS*, *CHI*, *F3H*, *DFR*, *ANS* and *UFGT* had significantly lower levels of transcription in A2 (black-purple with green calyx cultivar) and A4, (reddish-purple eggplant cultivar), thus contributing to lower flavonoids in A2 and A4. Generally, most of the structural genes involved in flavonoid biosynthesis were highly expressed in A1 and A3 with black-purple fruit peels [44]. This provides valuable data for further research on the molecular mechanism underlying anthocyanin accumulation in eggplant fruit peel. Further gene silencing of *SmGSTF1* exhibited bleaching around the injected fruit surface compared to the control (Figure 7); this suggests that *SmGSTF1* plays a crucial role in the purple color pigmentation process and hence could confer anthocyanin sequestration in eggplant fruit peel. In our analysis of *SmGSTF1* and ASGS-related genes, the expression was high in purple peel eggplant; therefore, distinctively the content of anthocyanin and expression of anthocyanin biosynthetic genes definitely would be suppressed in the silenced fruit. It is worth noting that *SmGSTF1* belongs to the same subclass (phi, F) as *CsGSTF1* known to be associated with anthocyanin hyperaccumulation in purple tea (*Camellia sinensis*) [42], *GhGSTF1* and *GhGSTF2* in cotton [61] and *RsGSTF12* in radish [62]. Similarly, *CmGSTF12* in *Camelina sativa* [63] and *PcGSTF12* in pear [64] have been found to influence anthocyanin sequestration in diverse plant tissues.

## 4. Materials and Methods

### 4.1. Identification and Characterization of GST Family Genes in Eggplant

For the comprehensive study of the eggplant *SmGST* genes, the Arabidopsis genome sequences served as the model plant for the search query using blast search on the eggplant genome database (http://eggplant-hq.cn./, accessed on 20 October 2023). Subsequently, sequences and annotations of the eggplant *SmGST* genes were obtained. A total of 64 GST family genes and their corresponding sequences in Arabidopsis were acquired from the TAIR database (accessible at: https://www.arabidopsis.org/, accessed on 20 October 2023). The GST genes were further subjected to a reliability test by using two methods; namely, blast and hmm search. Subsequently, to find *SmGST* proteins, the GST-C domain from the Pfam database; PF00043 (http://pfam.xfam.org/, accessed on 20 October 2023) was utilized in searching for the Hidden Markov model (HMM) in the genome files obtained from the eggplant genome (http://eggplant-hq.cn./, accessed on 20 October 2023).

The conserved domain database which is accessible via the link (http://www.ncbi.nlm.nih.gov/Structure/cdd/wrpsb.cgi?, accessed on 20 October 2023) yielded the domain of SmGST proteins. To denote the identified proteins, the prefix “Sm” for *Solanum melongena* preceded by a subfamily identifier was applied (such as *SmGSTU*, *SmGSTF*, *SmGSTT*, *SmGSTZ*, *SmGSTL*, *SmTCHQD*, *SmDHAR*, *SmEF1Bγ*, *SmMGST* and *SmGHR)* representing the respective subfamilies and an ascending number for the respective gene (such as *SmGSTU*1), as described previously [30]. To determine the length of the amino acid, theoretical isoelectric point (PI) and molecular weight of protein sequences were submitted to the protoparam tool, which is available online (https://web.expasy.org/protparam/, accessed on 20 October 2023). The subcellular localizations of SmGST proteins were predicted using the CELLO version.2.5 (http://cello.life.nctu.edu.tw/, accessed on 20 October 2023) and WOLF-PSORT (https://wolfpsort.hgc.jp/, accessed on 20 October 2023) online tools.

### 4.2. Phylogenetic, Motif and Protein Structure Analyses of GST Genes

Phylogenetic analysis was conducted to aid in identifying the relationship among species GST proteins. The GST protein sequences from the eggplant genome (http://eggplant-hq.cn./, accessed on 20 October 2023), tomato genome database (https://solgenomics.net/, accessed on 20 October 2023) and Arabidopsis (https://www.arabidopsis.org/, accessed on 20 October 2023) were utilized to obtain the molecular phylogeny using MEGA-X with the neighbor-joining (NJ) method and 1000 bootstrap replicates with the default settings [65]. GST protein sequence alignment for the phylogenetic tree was achieved by ClustalW [66] and saved in Newick format for further modification. Decoration of the phylogeny was accomplished using iTOL (https://itol.embl.de/itol.cgi/, accessed on 20 October 2023) [67]. Gene architectural analysis was conducted in Tb tools following the genomic sequences of eggplant to give accounts for exon and intron organizations of *SmGST* genes. The domains and motifs of *SmGST* genes were also analyzed in TBtools [68] and Weblogo3 (http://weblogo.threeplusone.com/create.cgi, accessed on 20 October 2023) was applied to enhance the sequences. Conserved protein domains of the *SmGST* were generated by MEME (https://meme-suite.org/meme/tools/meme, accessed on 20 October 2023) [69].

### 4.3. Chromosomal Localization, Gene Duplication and Collinearity Analyses of GST Genes

The 40 *SmGST* gene chromosomal locations were plotted through TBtools [68] following the information on *GST* genes obtained in the eggplant genomic file. The collinear relationships of *SmGST* in the eggplant genome were then analyzed using the default parameters of MCScanX. Similarly, the collinearity of the relationships among the *GST* genes in eggplant, Arabidopsis and tomato were conducted through the Multiple Collinearity Scan toolkit (MCScanX) and aligned by dual synteny analysis with the setting maintained at default states [70]. Two major forms of gene duplication are tandem and segmental duplications, with the tandem comprising two or more chromosome-bound genes while the segmental consists of genes on different chromosomes. The syntenic relationship was visualized in the form of a circle plot using TBtools [68]. The occurrence of gene duplication event in terms of million years was computed following the formulae: T = Ks/2λ × 10^−6^, where λ = 1.5 × 10^−8^ synonymous substitutions in each site per annum for dicotyledons [71].

### 4.4. Analysis of Cis-Acting Element of SmGST Genes

The *SmGST* promoter sequences consisting of 2000 bp sequences upstream of the transcription start site were obtained from the genomic sequence. An online tool for cis-regulatory analysis known as PlantCARE (https://bioinformatics.psb.ugent.be/webtools/plantcare/html/, accessed on 21 October 2023) was used to determine *cis*-acting regulatory elements and further the data presented in a graph using Excel.

### 4.5. Expression Analysis Using Transcriptomic Data

The expression data of eggplant *SmGST* genes among eggplant varieties of different peel colors were obtained from our recent research group RNA-seq data (PRINA868105) [44]. Briefly, the total RNA was extracted from fruit peels of the five varieties using the RNeasy Plant Mini Kit (Qiagen, Hilden, Germany). The RNA quantity, integrity and purity were analyzed using a Nanodrop and an Agilent Bioanalyzer 2100 (Agilent Technologies, Palo Alto, CA, USA). The sequencing of the libraries was performed on an Illumina HiSeq platform. The clean reads for each library were 6.38–7.45 Gb after filtration. All clean reads were mapped to the eggplant reference genome, (http://eggplant-hq.cn, accessed on 21 June 2021) with match ratios of 96.64–97.91%. Normalization was performed by Fragments Per Kilobase of transcript per Million mapped reads (FPKM): These methods normalize the read counts by the length of the transcript and the total number of mapped reads. The eggplant varieties of different peel color phenotypes that were used for RNA-seq analysis comprised: A1 (Suqie 6 cultivar-black-purple with purple calyx), A2 (Suqie 801 cultivar green), A3 (Bulita cultivar-black purple with green calyx), A4 (Suqie 11 cultivar-white) and A5 (Hangqie 1 cultivar-reddish). The variety EP 26 (dark-purple fruit) and EP28 (light-purple cultivar) were used for the qRT-PCR analysis of the expression in the selected tissues. The samples were grown in a greenhouse at the Liuhe experimental station of the Jiangsu Academy of Agricultural Sciences. Fruit peel samples were collected 25 days post anthesis. The growth conditions and sampling of fruit tissues are well illustrated in the previous report [44].

### 4.6. Total Anthocyanin Analysis

A total of 100 mg of peel samples were ground finely and used for the extraction of anthocyanin. The solution consists of 2 mL of a buffer with a pH of 1.0, which contains 50 mM potassium chloride (KCl) and 150 mM hydrochloric acid (HCl). Additionally, a pH 4.5 buffer comprising 400 nM of sodium acetate and 240 mM of hydrochloric acid (HCL) was added. Vertexing of the extractant was done and subsequently stored for further analysis at 4 °C for 12 h. Afterward, the solutions were centrifuged at a speed of 10,625 force of gravity at a temperature of 4 °C for 20 min. The absorption was measured at a wavelength of 510 nm, using a spectrophotometer and the cumulative anthocyanin concentration was computed following the previously described formulae [72].

### 4.7. Vector Construction and Transformation of Eggplant

The tobacco rattle virus (TRV) system was employed to silence the *SmGSTF1* gene through virus-induced gene silencing [73]. The 300-base pair finest target regions of *SmGSTF1* were determined using the SGN VIGS Tool (https://vigs.solgenomics.net/, accessed on 5 October 2022) that facilitated cloning of specific primer into a pTRV2 vector. Subsequently, Agrobacterium tumefaciens GV3101 strain was transformed with the TRV1, empty TRV2 and TRV2-*SmGSTF1* plasmids. Before infiltration, proportionate volumes of the cultivation of the Agrobacterium TRV1 culture were infused with void TRV2 and a similar procedure was applied to TRV2-*SmGSTF1* cultures. The eggplant peels were subsequently infused with the mixtures using syringes without needles attached to the nozzle and applied underneath the calyx and the calyx was removed afterward. The eggplant fruits were placed immediately into the dark chamber at a temperature of 22 °C for 24 h and then transferred to normal light conditions for 5 days. Each combination comprised three biological replicates.

### 4.8. Quantitative Real-Time PCR Analysis

The RNA plant Extraction Kit (Tiangen, Beijing, China) was used for the extraction of the total RNA and cDNA synthesized using a reverse transcription kit according to the manufacturer’s protocols (TaKaRa, Shiga, Japan). Designing of primer pairs of eggplant *SmGATA* genes was executed using Primer3plus (Primer3Plus—Pick Primers). UltraSYBR mixture (High Rox) kit (ComWin Biotech Co., Ltd., Beijing, China) was used for qRT-PCR analysis with the settings consisting of 40 cycles for 15 s at 95 °C and 40 s at 60 °C on the real-time PCR system. Quantification of the expression was performed following the 2^−ΔΔCT^ method [74]. The Actin gene was used as an internal control.

### 4.9. Statistical Analysis

The quantitative experiments were conducted in three replications for accuracy and reliability. The data were analyzed using Microsoft Excel 2019. Statistical variation was performed using a *t*-test (*p* < 0.05). (via SAS software version 9.4 (SAS Institute Inc. Cary, NC, USA).

## 5. Conclusions

A genome-wide analysis of *GST* genes was conducted in the eggplant, resulting in the identification of 40 *SmGST* which can be classified into seven subgroups. The physicochemical characteristics and subcellular localization were also provided. The *SmGST* that were grouped in the genealogical tree of evolutionary relationship exhibited similar exon–intron architecture and motif composition. These 40 *SmGST* genes are spread unevenly across 10 chromosomes. By analyzing the collinearity of GST genes with other plants, we obtained several eggplant *SmGST* genes that have a collinearity relationship with those of Arabidopsis and tomato, depicting similar evolutionary characteristics of *SmGST* to such species. The expression of the *SmGST* varied significantly across different tissues and varieties. Notably, this study revealed that *SmGST* can orchestrate anthocyanin synthesis and sequestration, thus laying the groundwork for further research on the specific regulatory network for future research on the special regulatory machinery network of GST involved in anthocyanin biosynthesis and accumulation. The results presented provide novel insights into the biological functions of the *SmGST* in purple eggplant anthocyanin accumulation.

## Figures and Tables

**Figure 1 ijms-25-04260-f001:**
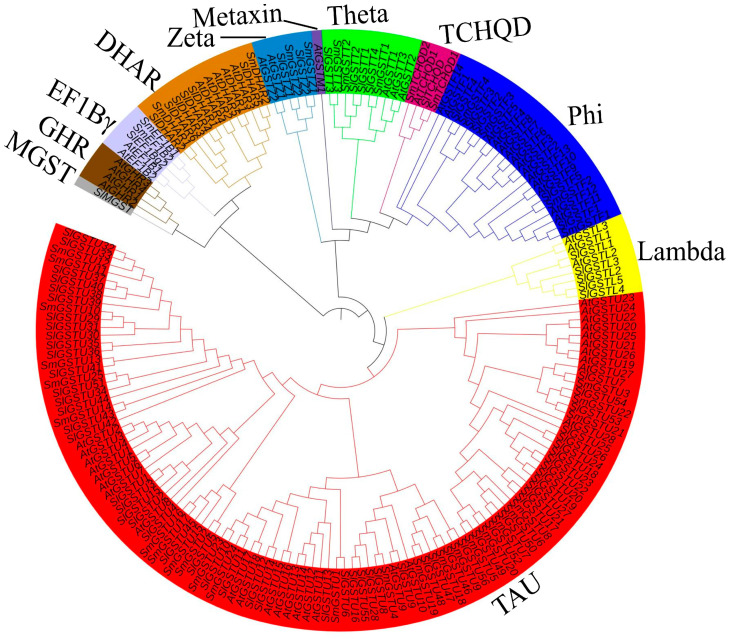
Phylogenetic tree of the 185 GST protein sequences from Arabidopsis, eggplant and tomato. Different subgroups—Tau, Phi, Lambda, TCHQD, Theta, Maxin, Zeta, DHAR, EF1Bγ, GHR and MGST—are designated with different colors.

**Figure 2 ijms-25-04260-f002:**
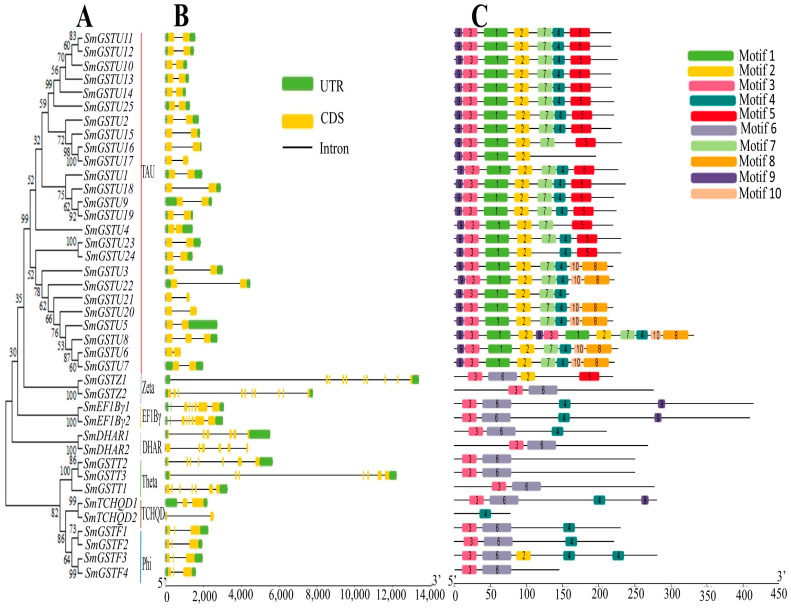
Phylogenetic relationships, gene structures of *SmGST* and conserved motifs of *SmGST* genes. (**A**) The phylogenetic tree of 40 *SmGST* family genes. *SmGST* subfamilies were assigned unique colors for identification. (**B**) Exon and intron structure of *SmGST* genes. Yellow rectangles are exons, while black lines are introns and green rectangles represent untranslated regions. (**C**) Different rectangular-shaped colors show the conserved motif patterns.

**Figure 3 ijms-25-04260-f003:**
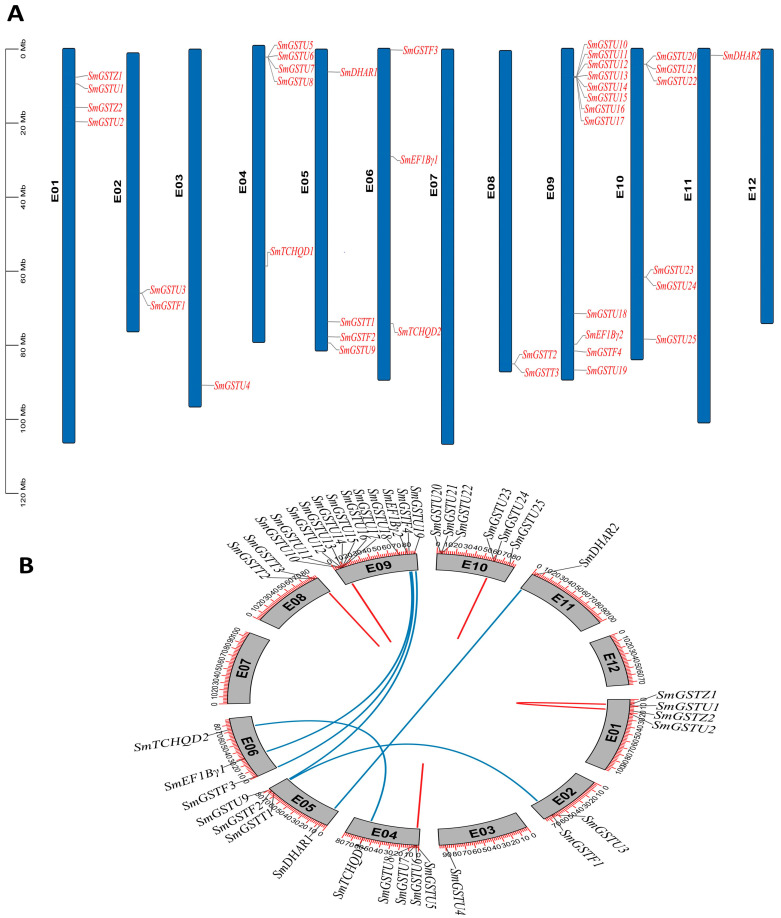
Chromosomal distribution and duplication of *SmGST*. (**A**) Representation of *SmGST* distributions on the chromosomes. Blue vertical bars are the chromosomes and their number is indicated to the left (black) while the gene number (red) is indicated to the right of each chromosome. (**B**) Illustration of the inter-chromosomal connections of *SmGST*. The blue lines denote segmental duplications while the red lines show tandem duplications of *SmGST* gene pairs.

**Figure 4 ijms-25-04260-f004:**
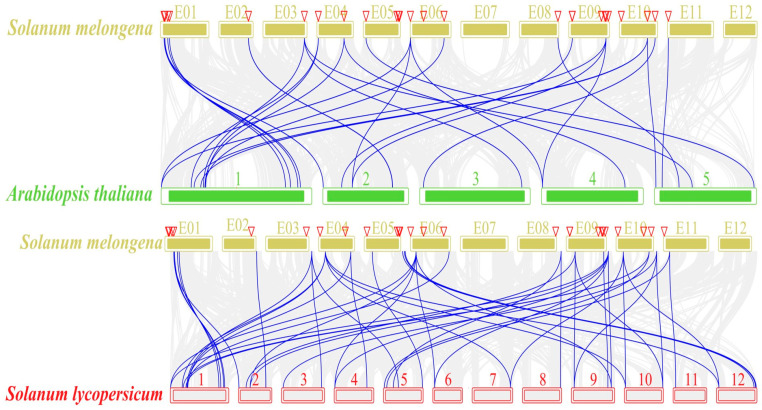
The collinearity analysis of *GST* genes among eggplant, Arabidopsis and tomato. The gray lines in the background are collinearity blocks, whereas the blue lines show the collinearity of *GST* gene pairs. The red triangles represent corresponding collinear *SmGST* genes.

**Figure 5 ijms-25-04260-f005:**
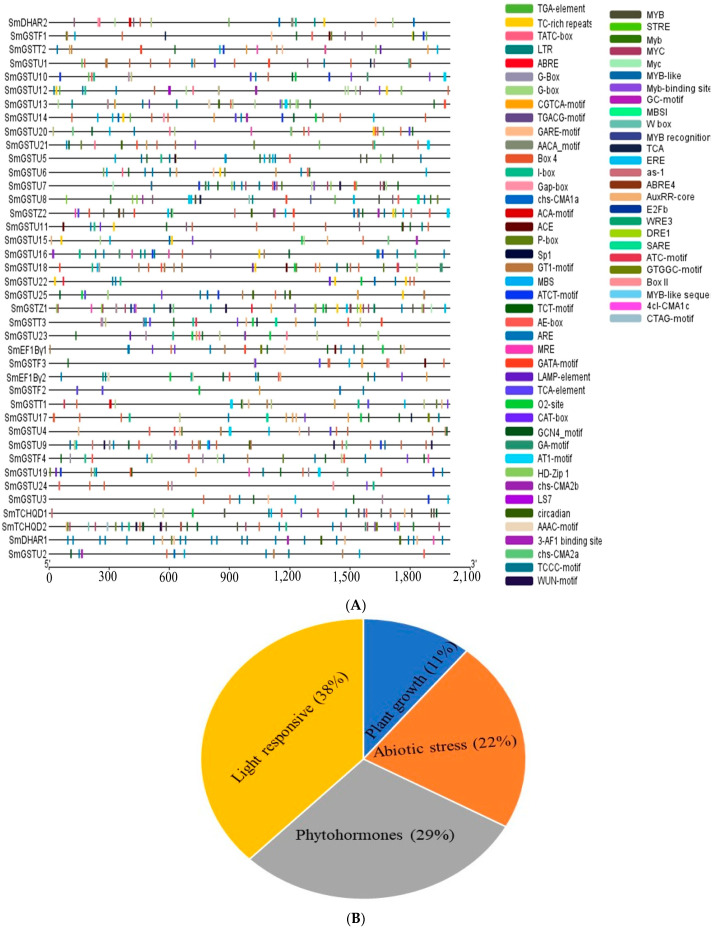
Promoter analysis of *SmGST*. (**A**) The main cis-acting elements in the promoter of *SmGST* genes. The cis-elements are represented by different colors. (**B**) The relative proportion of each group of promoter elements of light, plant growth, stress and phytohormones.

**Figure 6 ijms-25-04260-f006:**
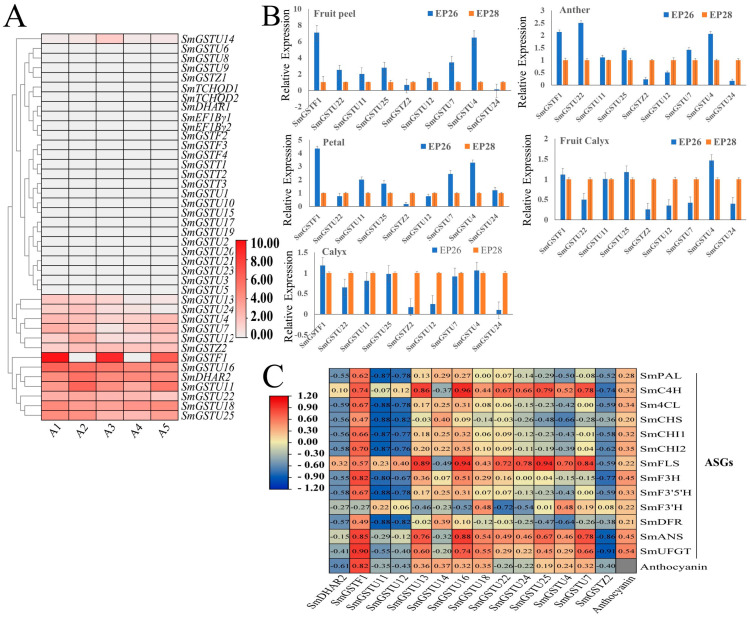
Expression profile of *SmGST* and anthocyanin content. (**A**) Expression profiles for *SmGST* genes in the peel of different eggplant varieties. FPKM values obtained from RNA-seq data and the levels of expression are plotted on Log 2-transformed (FPKM+1). Dark-purple cultivar; A1, green; A2, black-purple with green calyx cultivar; A3, white cultivar; A4 and reddish-purple cultivar; A5. (**B**) qRT-PCR expression patterns of 9 *SmGST* in the peel, anther, petal, fruit calyx and calyx, of EP26 and EP28 eggplant cultivars examined by qRT-PCR. Error bars denote standard error. (**C**) Correlation analysis of the expression profiles of *SmGST* genes, anthocyanin content and anthocyanin structural genes (ASGs) and relation with total anthocyanin content at *p* < 0.05. Expression profiles for *SmGST* genes obtained from RNA-seq data of the peel of different eggplant varieties. PAL (*Smechr0500713*), phenylalanine ammonia-lyase; C4H (*Smechr0603018*), cinnamate 4-hydroxylase; 4CL (*Smechr0302347*), 4-coumarate-CoA ligase; CHS (*Smechr0500409*), chalcone synthase; CHI (*Smechr0500261*; *Smechr1001863*), chalcone isomerase; F3H (*Smechr0202240*), flavanone 3-hydroxyl enzyme; FLS (*Smechr1001394*), flavonol synthase; DFR (*Smechr0202337*), dihydroflavonol reductase; ANS (*Smechr1002343*), anthocyanin synthase; UFGT (*Smechr0502047*), flavonoid 3-O-glucosyl transferase.

**Figure 7 ijms-25-04260-f007:**
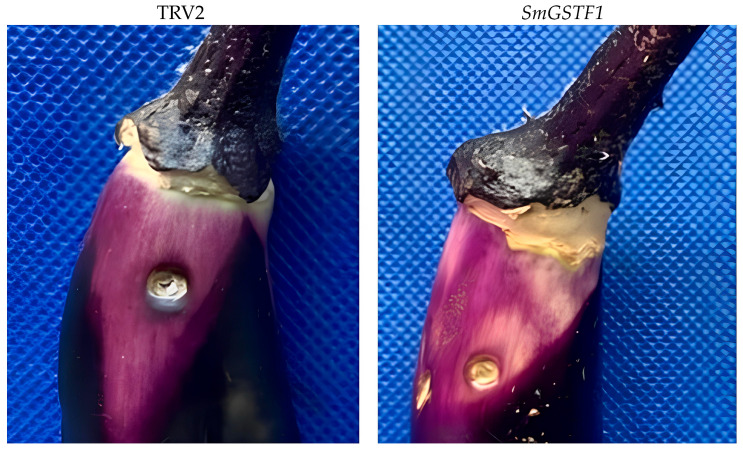
The effect of *SmGSTF1* silencing in VIGS eggplant fruits. Peel color change of eggplant fruit peels infused with Agrobacterium cells (TRV2; *SmGSTF1-TRV2*). A devoid TRV vector was the control. The injection was done underneath the calyx and the calyx was removed afterwards.

**Table 1 ijms-25-04260-t001:** Identification and characterization of *SmGST* genes.

Group	Gene Name	Gene Id	Chr	Start-End	Cds	AA	mW (kDA)	pI	Sub. Loc.
TAU	*SmGSTU1*	Smechr0101012	E01	9,617,214–9,619,141	1928	227	25.664	5.18	cytoplasm
	*SmGSTU2*	Smechr0101935	E01	19,836,231–19,837,973	1743	221	25.269	6.46	cytoplasm
	*SmGSTU3*	Smechr0201984	E02	64,921,863–64,924,878	3016	220	25.402	5.3	cytoplasm
	*SmGSTU4*	Smechr0303071	E03	90,785,487–90,786,908	1422	220	25.326	5.52	cytoplasm
	*SmGSTU5*	Smechr0400284	E04	3,173,220–3,175,948	2729	220	26.052	6.23	cytoplasm
	*SmGSTU6*	Smechr0400285	E04	3,176,679–3,177,460	782	227	26.695	5.33	cytoplasm
	*SmGSTU7*	Smechr0400287	E04	3,180,873–3,182,848	1976	222	25.764	5.91	cytoplasm
	*SmGSTU8*	Smechr0400288	E04	3,182,978–3,185,708	2731	332	38.905	5.67	cytoplasm
	*SmGSTU9*	Smechr0502567	E05	79,384,784–79,387,225	2442	221	25.895	5.51	chloroplast
	*SmGSTU10*	Smechr0900514	E09	7,607,904–7,609,024	1121	226	26.289	7.6	chloroplast
	*SmGSTU11*	Smechr0900520	E09	7,650,862–7,652,420	1559	217	25.204	5.59	cytoplasm
	*SmGSTU12*	Smechr0900521	E09	7,666,984–7,668,465	1482	217	25.287	5.17	chloroplast
	*SmGSTU13*	Smechr0900522	E09	7,693,565–7,694,619	1055	218	24.828	5.58	cytoplasm
	*SmGSTU14*	Smechr0900523	E09	7,693,565–7,694,619	1055	218	24.828	5.58	cytoplasm
	*SmGSTU15*	Smechr0900526	E09	7,849,254–7,851,059	1806	217	25.070	6.91	cytoplasm
	*SmGSTU16*	Smechr0900527	E09	7,866,824–7,868,709	1886	232	26.968	5.18	cytoplasm
	*SmGSTU17*	Smechr0900529	E09	7,890,677–7,891,855	1179	196	22.930	7.64	nucleus
	*SmGSTU18*	Smechr0901584	E09	71,621,469–71,624,378	2910	237	27.918	5.6	cytoplasm
	*SmGSTU19*	Smechr0902464	E09	86,899,097–86,900,533	1437	224	26.242	5.34	cytoplasm
	*SmGSTU20*	Smechr1000365	E10	4,291,432–4,293,058	1627	220	25.588	5.55	cytoplasm
	*SmGSTU21*	Smechr1000366	E10	4,296,416–4,297,666	1251	159	18.369	5.52	nucleus
	*SmGSTU22*	Smechr1000367	E10	4,304,841–4,309,315	4475	222	26.047	5.85	nucleus
	*SmGSTU23*	Smechr1001526	E10	61,779,912–61,781,751	1840	231	25.955	6.18	cytoplasm
	*SmGSTU24*	Smechr1001527	E10	61,782,434–61,783,844	1411	231	25.832	5.2	cytoplasm
	*SmGSTU25*	Smechr1002369	E10	78,540,197–78,541,469	1273	221	25.308	5.46	cytoplasm
Phi	*SmGSTF1*	Smechr0201996	E02	64,999,172–65,001,420	2249	230	26.330	5.87	chloroplast
	*SmGSTF2*	Smechr0502407	E05	77,720,819–77,722,753	1935	221	25.204	5.57	chloroplast
	*SmGSTF3*	Smechr0600022	E06	434,013–435,961	1949	281	31.636	7.81	chloroplast
	*SmGSTF4*	Smechr0902065	E09	81,756,012–81,757,597	1586	145	16.400	4.92	chloroplast
Theta	*SmGSTT1*	Smechr0502113	E05	73,646,479–73,649,739	3261	277	31.134	6.63	cytoplasm
	*SmGSTT2*	Smechr0802345	E08	84,625,796–84,631,454	5659	250	28.457	9.35	cytoplasm
	*SmGSTT3*	Smechr0802346	E08	84,631,786–84,644,013	12,228	250	28.676	9.45	cytoplasm
EF1Bγ	*SmEF1Bγ1*	Smechr0600694	E06	29,261,081–29,264,158	3078	414	47.144	5.93	cytoplasm
	*SmEF1Bγ2*	Smechr0901946	E09	79,912,280–79,915,304	3025	409	46.476	6.09	chloroplast
DHAR	*SmDHAR1*	Smechr0500548	E05	6,181,924–6,187,450	5527	211	23.544	6.6	cytoplasm
	*SmDHAR2*	Smechr1100162	E11	1,888,515–1,892,865	4351	268	30.058	6.14	chloroplast
TCHQD	*SmTCHQD1*	Smechr0401375	E04	59,614,117–59,616,319	2203	280	32.854	8.99	cytoplasm
	*SmTCHQD2*	Smechr0601694	E06	74,293,706–74,296,233	2528	78	9.166	4.4	chloroplast
Zeta	*SmGSTZ1*	Smechr0100825	E01	7,795,166–7,808,572	13,407	220	24.872	6.32	cytoplasm
	*SmGSTZ2*	Smechr0101620	E01	15,928,247–15,936,036	7790	276	31.585	6.61	chloroplast

Table legend: Chromosome (Chr), amino acids (AA), coding sequence (Cds), molecular weight in kilodaltons (mW, kDa), theoretical isoelectric points (pI), subcellular localization (Sub. loc.).

## Data Availability

All needed genome sequences and genome annotation files of eggplant were obtained from the eggplant genome (http://eggplant-hq.cn./, accessed on 22 October 2023) and the published GST sequences of *Arabidopsis thaliana* were acquired from the TAIR database (http://www.arabidopsis.org/, accessed on 22 October 2023) and tomato genome database (https://solgenomics.net/, accessed on 22 October 2023). The transcriptome sequencing data of different eggplant fruit peel tissues used in this study was obtained from a previous report RNA-Seq datasets presented and can be found in NCBI Sequence Read Archive (SRA) under the accession number PRINA868105 (https://www.ncbi.nlm.nih.gov/bioproject/?term=PRINA868105, accessed on 28 October 2023). All databases in this study are available to the public.

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
