# Peer review of "Genome-Wide Identification of Glutathione S-Transferase Genes in Eggplant (Solanum melongena L.) Reveals Their Potential Role in Anthocyanin Accumulation on the Fruit Peel"

_ijms, 2024, doi:10.3390/ijms25084260_

Round 1

Reviewer 1 Report (Previous Reviewer 3)

Comments and Suggestions for Authors

The question in Figure 6b. is the same...

As delta delta CT is used. the ‘Relative expression’ should be made. There should be one sample that has 1 value and quantified the fold expression change in other samples. 

In the case of "How are expression values normalized across the SRAs?" The author mentioned normalization for the single dataset, but when dealing with multiple datasets, normally, Z-score normalization is used if possible authors can follow it.

Overall manuscript has been improved.

Author Response

Thank you for taking your valuable time to review our manuscript and to improve its contents.

Kindly find the attached responses. Thank you so much.

Reviewer 2 Report (New Reviewer)

Comments and Suggestions for Authors

Page 8 line 238, change “promoter” to “promoters”; line 239, change “cis-acting”to “Cis-acting”

 Page 10 line 284, I guess “dark -the purple” may be “the dark-purple”; line 285, RT-qPCR may be qRT-PCR.

 Page 13 line 346-350, “Identification of GST genes in other plant species has been reported for example, 55 GSTs in Arabidopsis [46], 38 GSTs from the apple (Ma lus domestica [38], and 82 GSTs in radish (Raphanus sativus) [47], 90 in potato (Solanum tuberosum L.) [48], 90 in tomato [49], 74 in soybean (Glycine max) [50], potato [51], radish [52], pear (Pyrus communis L.) [43], sweet potato (Ipomoea batatas) [53].” The authors may lost some information for potato [51], radish [52], pear (Pyrus communis L.) [43], sweet potato (Ipomoea batatas) [53]. 

 Add reference for the “Transformation of Eggplant” in part 4.7.

 How were the transgenic eggplant fruits selected? Did the vector have a reporter gene? I also suggest the authors use qRT-PCR or others to detect the expressions of SmGSTF1 in the TRV2 and SmGSTF1-TRV2 eggplant fruits.

 The authors analyzed that SMGSTF1 gene might be involved in anthocyanin biosynthesis by gene silencing experiment. The results are interesting, but a phenotypic map alone is weak in proving its function. It is recommended to increase the measurement of anthocyanin content and other physiological data as well as the changes of the ASGs expressions affected by SmGSTF1-silencing

Author Response

Thank you for taking your valuable time to review our manuscript and to improve its contents.

Kindly find the attached responses. Thank you so much.

Reviewer 3 Report (New Reviewer)

Comments and Suggestions for Authors

Manuscript ijms-2941275 entitled "Genome-Wide Identification of Glutathione S-Transferase Genes in Eggplant (Solanum melongena L.) Reveals Their Potential Role in Anthocyanin Accumulation on the Fruit Peel” studies of the SmGST family genes and provides the foundation for deciphering molecular investigations into the functional analysis of SmGST genes in eggplant.

The topic fits within the scope of the journal and it is interesting. The manuscript expands existing knowledge as results obtained were expected and comparable to those obtained in other papers. It is a very complete manuscript, which shows A LOT of data and I believe that it deserves to be published in a high quality of journal like International Journal of Molecular Sciences. My suggestion is to accept the manuscript for publication after some minor changes are made.

Good job!

Suggestions:

1. The line 7 of Table 1, “Cytoplasm” should be change to “cytoplasm”.

2. 187-188. The title should be revised to “ (C) Yellow rectangles are exons, while black lines are introns, and green rectangles represent untranslated regions.”

3. line 239: The initial letter should be capitalized, please revised.

4. line 535: Where is the manufacturer of the SAS software, please added.

5. The format of the references needs to be modified. For example, the first letter of the article title in Ref.1, Ref.3, and Ref.4 is uppercase, while the first letter of the article in Ref.5, Ref.6, Ref.7, etc. is lowercase. Please modify it and unify the format of the entire reference section.

Author Response

Thank you for taking your valuable time to review our manuscript and to improve its contents.

Kindly find the attached responses. Thank you so much.

This manuscript is a resubmission of an earlier submission. The following is a list of the peer review reports and author responses from that submission.

Round 1

Reviewer 1 Report

Comments and Suggestions for Authors

As eggplant has a huge variability for fruits characteristics like shape, size, fruit color and other preferential traits. The color is considered to be the most important character for its preference. The study of GST Family Genes in the manuscript entitled "Genome-Wide Identification of Glutathione S-Transferase Genes in Eggplant Reveals their Potential role in Anthocyanin accumulation on the Fruit Peel" presents a thorough analysis of forty GST genes in the eggplant genome and further its characterization by suppression of SmGSTF1 which modulates the anthocyanin accumulation in peels, but improving the highlighted points could further strengthen its scientific rigor and readability.

The manuscript presents a high level of scientific and technical temperament but in the case of anthocyanin quantification, normal spectrophotometric methods have been used. Linking the quantitative data of anthocyanin with respective GST family genes will enhance the quality of manuscript. 

One illustration representing Phylogenetic analysis of GST family genes and phenotypic diversity can be included.

The gene silencing experiment was performed in harvested fruits under the calyx region. Have the authors check the same in the fruit bearing plant itself.

Author Response

Thank you very much for your valuable comments and suggestions

Reviewer 2 Report

Comments and Suggestions for Authors

Dear Authors,

The focus of the manuscript entitle “Genome-Wide Identification of Glutathione S-Transferase 2 Genes in Eggplant Reveals their Potential Role in Anthocyanin 3 Accumulation on the Fruit Peel“ is  identification and characterization of GST genes from  Solanum melongena L. The authors performed a comprehensive analysis of the S. melongena L. genome and obtained a total of 40 GST members. Their genetic composition, phylogenetic relations, and expression patterns in fruit tissues were in silico analyzed. By using virus induced gene silencing (VIGS), the role of SmGSTF1 gene was explored in anthocyanin sequestration in eggplant purple fruit.

The presented methods are original. It is great amount of work with scientific relevance. The study presents novel information about identification and characterization of GSTs in eggplant which remains unknown.

 I find some minor flaws and weaknesses in the manuscript:

-        the authors need to provide the Latin name of eggplant (Solanum melongena L) as well, the first time when they mention it in the text (in the abstract) – line 17. The Latin names give more scientific relevance.

-        In my opinion “Solanum melongena L should replace “eggplant” in the title. But depends on the authors' preferences

-        In line 19, 89, 97, 143, 168, 170, 173, 177 etc….. – check for missing comma character, full stop or Capital letter

-        Mention the Latin name of the plants when you use it for the first time in the text – line 27

-         All of the gene’s names should be in Italic - line 31, line 133, line 163

-        the text below the Table 1 should be specified as a legend

-        line 147 – delete “subfamily” (it is additional word)

-        line 333 – delete Islam et al., 2018

-        in the titles of M&M there are additional comma characters before the word “and”

English language is fine. Figures with good quality and informative. The references are appropriate.

I recommend minor revision.

Author Response

Thank you very much for your valuable suggestions and comments. We are looking forward to your positive response

Reviewer 3 Report

Comments and Suggestions for Authors

Gene name should be in italics, and protein name normal. Please check throughout the manuscript.

Line 19-20: grouping of GST genes ..based on the Arabidopsis thaliana "evolutionary relationship" check for term...evolutionary relationship... does it mean eggplant relationship with Arabidopsis?

Line 20: The 40 genes associated with GST.....they are GST genes not associated.

Line 28-29: Rewrite RNA-seq data analysis...for clarity.

Figure 2C. Provide details of the motif sequence and functional relevance in the supplementary figure.

Section 2.6: its general representation of data.. what is the difference among the family members? Provide a detailed table with supplementary data (for better presentation authors can follow this paper:doi: 10.3389/fpls.2023.1216082).

Line 296-296: An online SRplot to...shift to MM.

-How are RNA-Seq data analyzed? How are expression values normalized across the SRAs?

Figure 6b. This is not a ‘Relative expression’ because there is nothing to make a relative comparison. Please explain which units are used for ‘Expression level’ in the Y-axis and how they were calculated.

Figure 6c: Which expression profiles are used: RNA-seq, real-time PCR, and organ?

Figure 7: Authors can quantify anthocyanin content/gene expression in VIGS eggplant fruits to validate their findings.

Comments on the Quality of English Language

Moderate editing of English language required.

There are many typographical errors, including full stop, italics apart from language clarity

Author Response

Thank you very much for your valuable suggestions and comments. We are looking forward to your positive response. Thank you very much 
